# Road-Scene Parsing Based on Attentional Prototype-Matching

**DOI:** 10.3390/s22166159

**Published:** 2022-08-17

**Authors:** Xiaoyu Chen, Chuan Wang, Jun Lu, Lianfa Bai, Jing Han

**Affiliations:** Jiangsu Key Laboratory of Spectral Imaging and Intelligent Sense, Nanjing University of Science and Technology, Nanjing 210094, China

**Keywords:** intelligent vehicles, scene-parsing, prototype learning, attention mechanism

## Abstract

Road-scene parsing is complex and changeable; the interferences in the background destroy the visual structure in the image data, increasing the difficulty of target detection. The key to addressing road-scene parsing is to amplify the feature differences between the targets, as well as those between the targets and the background. This paper proposes a novel scene-parsing network, Attentional Prototype-Matching Network (APMNet), to segment targets by matching candidate features with target prototypes regressed from labeled road-scene data. To obtain reliable target prototypes, we designed the Sample-Selection and the Class-Repellence Algorithm in the prototype-regression progress. Also, we built the class-to-class and target-to-background attention mechanisms to increase feature recognizability based on the target’s visual characteristics and spatial-target distribution. Experiments conducted on two road-scene datasets, CamVid and Cityscapes, demonstrate that our approach effectively improves the representation of targets and achieves impressive results compared with other approaches.

## 1. Introduction

Road-scene parsing has always been a challenging task in the area of computer vision: a complex background will destroy the visual structure of the targets with occlusion, motion blur, illumination, and scale variation. At the same time, different classes may also show visual characteristics similar to the targets under poor imaging conditions, which makes target recognition difficult [1].

Currently, semantic segmentation algorithms are mainly based on the Fully Convolutional Network (FCN) structure, which consists of a backbone network for feature extraction and a classifier for pixel-level classification [2]. In the FCN models, the performance of semantic segmentation as a pixel-level classification process relies on features output by the backbone and the network optimization because of the complex road scenes and the changeable targets. So, FCN-based models enrich the feature representation via a complex, network-structure design, and they improve the optimization strategy for better performance, including: (1) combining local and global information for multi-scale feature-fusion [3,4]; (2) adopting a larger convolution kernel to enhance the receptive field [5,6]; (3) designing an encoder–decoder network structure to enhance different levels of features [7,8]; and (4) using an auxiliary-loss function for better optimization [9,10]. These practices enrich receptive information for the features extracted by the network, but they ignore the relationships and differences between the targets and the background.

In addition, the widely used loss-functions in the semantic segmentation task, such as cross-entropy loss, are based on pixels, and the annotation labels of different points provide supervised clues independently, resulting in poor structural consistency in the segmentation results. The main ideas for making full use of this structural information are to use post-processing and to add a structural-loss function. Conditional random field (CRF) is a post-processing method which measures the spatial similarity for pixel-level prediction correction [11]. Generative adversarial networks (GANs) use the trained discriminator to discriminate whether the prediction results are consistent with the labeled data at a structural level to enhance the integrity of the target. However, as the models become more complicated, the optimization process is prone to instability and collapse [12]. AAF proposes the concept of an adaptive affinity field to construct the local structural-loss function and extract structural information [13]. However, the receptive field in the adaptive affinity field is located at the pixel level, and the improvement is limited by the small receptive field.

The Nearest Class Mean (NCM) classifier is a classical and promising classification model for multiclassification tasks, in which the target is classified according to the distance between the sample representations and the class-mean representations [14]. The class means, also called the “class prototypes”, represent the class feature-center. NCM recognizes samples by analyzing global class-samples, which enhance the spatial consistency. NCM is different from the widely used Softmax classifier because Softmax analyses the weight vectors of all of the classes and calculates the classification score. In contrast, the class mean in NCM is dependent, and the difference between classes is maintained, which is easily adapted to various scenes. However, NCM is a weak, linear classification and cannot deal with complex, high-dimension features. LDN has further developed Deep Nearest Class Mean (DeepNCM), which directly learns a highly nonlinear, deep representation and approximates the class means with online estimations [15]. DeepNCM allows the class means to closely follow the drifting representation, and the visual representation can be well-generalized to new classes.

NCM series’ methods construct classifications based on the class means (class prototypes), which represent the class feature-center, so the samples are classified based on the similarities to the class means. However, the class means ignore the differences between class samples in complex road scenes; the difficult and easy examples contribute equally, which leads to a bias in the estimation of the class means. Besides, the similar class means from different classes increase the false classification. So, the inter- and intra- sample differences need to be considered in the class-mean estimation.

The recently developed attention mechanisms in computer vision tasks have achieved great success, constructing spatial relations and improving the integrity of feature structures. In complex road scenes, similar samples from different classes are hard to distinguish, and targets with interference are challenging to recognize. So, it is necessary to enhance the sample features in terms of the class characteristics and the background in road scenes.

This paper proposes Attentional Prototype-Matching Network (APMNet) for high-quality road-scene parsing. Different from end-to-end semantic segmentation models, APMNet reorganizes the segmentation process into two steps: class-prototype regression and attentional prototype-matching. In the class-prototype regression process, we used a large amount of data to learn various prototype features, and then we used the similarity between target features and class-prototype features as the classification basis. In order to enhance the separability of features, we used spatial- and class-attention feature-enhancement methods to enhance the segmentation performance of the network. Compared with DeepNCM method, APMNet abandons the scheme of using the class-feature mean as the measurement and discrimination basis, directly regressing the class prototype as the measurement and discrimination basis in the training process so as to increase the flexibility of the network and improve the segmentation accuracy. In addition, the attention-feature-enhancement module enhances the structural integrity and accuracy of the prediction results by strengthening the interaction between the class prototype and the input scene.

The highlights of this work are listed as follows:We have improved the NCM framework, and we propose the novel APMNet to segment targets by matching candidate features with target prototypes regressed from labeled road-scene data for robust recognition.We have designed the Sample-Selection and the Class-Repellence Algorithms in the prototype-regression progress to obtain reliable target prototypes.We have built the class-to-class and target-to-background attention features to increase feature recognizability based on the targets’ visual characteristics and spatial-target distribution.

## 2. Related Work

### 2.1. Classification in Semantic Segmentation

A semantic segmentation task is typically formulated as a pixel-wise classification, and almost all FCN-like segmentation models comprise an encoder part and a classifier. Many FCN-like models enhance their performance by boosting the perception field and aggregating the long-range context in the encoder. PSPNet exploits the capability of global context information via different-region-based context aggregation through the pyramid-pooling module and proposes a pyramid scene-parsing network for feature enhancement [9]. ASPP adopts atrous convolutions to build a spatial pyramid structure for perception-field boosting [16]. BiSeNet introduces a two-branch network to obtain rich spatial information and sizeable receptive-field extraction [17].

However, studies have demonstrated that simply focusing on pixel-wise supervision is not enough, and local structures are still essential for high-quality segmentation. AAF captures and matches the semantic interactions among neighboring pixels based on labels, and it verifies the local structure during the training phase [13]. Gated-SCNN adopts a two-stream framework to connect shape information to obtain high-quality boundaries [18]. CFNet estimates the distribution of co-occurrent features for fine-grained representations [19].

Despite the fact that the semantic segmentation models are differently designed than FCN-based frameworks, the models can be placed in one category by considering the Softmax parameters as learnable class prototypes. The prototypes are simply regressed in a parametric manner, and the representative ability is limited.

### 2.2. Prototype-Based Models for Recognition

The prototype-based methods introduce metric leaning to the recognition tasks. DeepNCM cumulates the mean of the features of the classes’ samples to transform a classification to a target-matching task [15]. Tianfei Zhou et al. uncovered the limitations of parametric segmentation models and proposed non-parametric models in light of a prototype view [20]. Pascal Mettes et al. adopted hyperspheres as output spaces and position prototypes through data-independent optimization [21]. Xiaoyu Chen et al. proposed multi-view convolution to replace parametric Softmax in supervised, semantic segmentation [22]. Loic Landrieu et al. integrated a prototypical network in the supervision to model the hierarchical class-framework [23].

The prototype-based models are promising, but in the existing methods, prototypes are trained or defined prior to training, and complex backgrounds in complex environments interfere with the matching process.

### 2.3. Attention Mechanism in Semantic Segmentation

The recently developed attention mechanisms in segmentation networks have been designed to improve performance. DANet is a dual-attention network that has been proposed to adaptively combine local and global dependencies [24]. CCNet is a Criss-Cross Network that proposes to exploit contextual information effectively and efficiently [25]. RANet configures the information pathways between the pixels in different regions for better contextual representations [26].

In this paper, we have referenced the existing attention-mechanism methods, and we have built a class-to-class attention and target-to-background attention based on prototype-matching for high-performance semantic segmentation.

## 3. Method

This section describes the design of our scene-parsing network based on our Attentional Prototype-Matching (APMNet). The prototypes in APMNet are defined as the features that have the maximum similarity with samples of a specific class. APMNet consists of the prototype-based feature-extractor, the attentional-enhancement module, and the predictor. The predictor is a sequence of convolutional layers used to convert features to segmentations. In the classic semantic-segmentation network, the feature-extractor extracts the deep features of a given input image with multi-scale and multi-receptive fields, and then it obtains the results directly through the predictor. Aimed at the interference problem in complex road scenes, APMNet designs the class attention module to analyze the global and local target-distribution according to the different road parts, and it enhances the discrimination ability of features for better segmentation performance.

Figure 1 shows the two main procedures of APMNet; the dotted line indicates the Class-Prototype Regression, and the cyclical arrow indicates the Attentional Feature-Enhancement in the training phase. The lower part shows the testing flow-path after training.

Given the input image of I, compute the feature $f_{i,j}$ of the coordinate $\left(i,j\right)$ by the extractor. Then, build the relation matrix M with the class-prototype feature proto and the feature map $f_{i,j}$ for feature attentional-enhancement. Finally, estimate the segmentation results by the predictor.

### 3.1. Classification in Semantic Segmentation

The class-prototype feature is the basis of constructing the relationship matrix, which is regressed from a large number of labeled images. First, the class-prototype feature proto is defined as:(1)$$proto=\left[p_{1},\;p_{2},\;\ldots ,\;p_{N}\right],n\in \left[0,\;N-1\right]$$
where the n indicates the class number, and the $p_{class}$ is the prototype feature of the specific class. Initialize the proto according to the pixel-level labels $gt_{i,j}\in \left[0,N-1\right]$:(2)$$P_{n}=\frac{\sum_{i=1}^{H}\sum_{j=1}^{W}(\left(n==gt_{i,j}\right)\cdot f_{i,j})}{\sum_{i=1}^{H}\sum_{j=1}^{W}(\left(n==gt_{i,j}\right)\cdot 1)}$$

Compute the feature distances $D_{i,j}$ between $f_{i,j}$ and $P_{gt}$, and upgrade the $P_{n}$ through the minimization of $D_{i,j}$ to make the class-prototype feature proto more accurate. The prototypes are regressed, while the $f_{i,j}$ are fixed, and the regression process is defined as:(3)$$\min_{P}D_{i,j}=dist\left(f_{i,j},P_{gt}\right)$$
where $dist\left(\cdot \right)$ indicates the distance metric, and in this paper, we use the Euclidean distance for its simplicity and effectiveness. In the regression loop, we designed the Sample-Selection Algorithm and the Class-Repellence Algorithm for efficient prototype regression.

#### 3.1.1. Sample-Selection Algorithm

In the prototype regression, we adopted the Sample-Selection Algorithm (SSA) to select hard samples for regression. The SSA loss is defined as:(4)$$loss_{SSA}=\sum sReLU\left(D_{i,j}\right)$$
(5)$$sReLU\left(D_{i,j}\right)=\left\{\begin{array}{c} 0\\ D_{i,j}-\varepsilon \end{array}\right.\;\;\begin{array}{c} D_{i,j}\leq \varepsilon \\ D_{i,j}>\varepsilon \end{array}$$
where ε is the threshold to determine whether $D_{i,j}$ is hard sample, and sReLU is used as an activation function to reduce easy samples in the prototype regression.

Considering the complexity of the target and the background interference in the driving scenarios, we proposed the Class-Repellence Algorithm (CPA) to amplify the difference between the prototype-feature $P_{class}$. Specifically, the target or interference features of different classes with similar visual appearances have similar numerical distributions, which will lead to misclassification. To solve this problem, this section proposes the Class-Repellence Algorithm, which realizes the feature-isolation of class prototypes by punishing the similarity between class prototypes, so as to improve the difference between each $P_{class}$. The CPA loss is defined as:(6)$$loss_{CPA}=\sum dist\left(p_{m},\;p_{n}\right),\;m\neq n$$

Finally, the total loss-function of the Class-Prototype Regression is represented as:(7)$$loss_{p}=loss_{loss}+\alpha \;loss_{CPA}$$
where α indicates the hyperparameters to improve the training, which is set according to the experiments.

### 3.2. Attentional Feature-Enhancement

This section demonstrates the proposed attentional-enhancement module (AFE). The AFE constructs mutual relationships between prototypes and candidate features, and it implements spatial and specific class-enhancement based on the mutual relationships. The mutual relationships $D_{i,j}^{n}$ are formulated as
(8)$$D_{i,j}^{n}=Sim\left(f_{i,j},P_{n}\right)$$
where $D_{i,j}^{n}$ indicates the similarity value between the feature $f_{i,j}$ and the n-th class-prototype feature at the coordinate $\left(i,j\right)$. Sim() indicates the similarity computation; in this paper, we have adopted a cosine similarity-metric. $D_{i,j}^{n}$ indicates the possibility that the feature $f_{i,j}$ is classified to class n. The features with strong characteristics show a high level of similarity with specific class prototypes, and ambiguous features need to be further distinguished according to the relationships between spatial and class prototypes. This paper has exploited the spatial and characteristic features with the relationships of class-to-class and target-to-background.

#### 3.2.1. Class-to-Class Attention for Feature-Enhancement

The motivation for the class-to-class attention for feature-enhancement is the reconstruction of the candidate features based on the mutual relationships $D_{i,j}^{n}$. In the reconstruct, the prototypes with high similarity values play the main role, and conversely, the less similar prototypes are less important.

First, we implemented the normalization according to the prototype dimension in the relationship matrix $D_{i,j}^{n}$ to reduce the scale difference of similarity values among the prototypes:(9)$$C_{i,j}^{n}=\frac{\exp\left(D_{i,j}^{n}\right)}{\sum_{n=1}^{N}\exp\left(D_{i,j}^{n}\right)}$$
where $C_{i,j}^{n}$ represents the normalized similarity score of the candidate feature $f_{i,j}$ to the prototypes $\left[p_{1},\;p_{2},\;\ldots ,\;p_{n}\right]$. Then, the attentional reconstruction of candidate features is computed as:(10)$$f_{i,j}^{C2C}=\left[C_{i,j}^{1},\;C_{i,j}^{2},\;\ldots ,\;C_{i,j}^{3}\right]{\left[p_{1},\;p_{2},\;\ldots ,\;p_{n}\right]}^{T}$$
where $f_{i,j}^{\prime }$ indicates the enhanced feature, which combines the spatial textures and prototype characteristics and amplifies the intra-class difference for fine analyses.

#### 3.2.2. Target-to-Background Attention for Feature-Enhancement

The target-to-background attention focuses on strong interferences in the background, and the motivation is to refine the candidate features to reduce the background interference during the feature enhancement. In order to reduce the spatial difference, normalization according to spatial dimension is implemented:(11)$$S_{i,j}^{n}=\frac{\exp\left(D_{i,j}^{n}\right)}{\sum_{i=1}^{H}\sum_{j=1}^{W}\exp\left(D_{i,j}^{n}\right)}$$
where $S_{i,j}^{n}$ indicates the normalized scores of the prototype according to the spatial dimension. So, features with high scores show a high possibility of being targets, and features with lower scores are more likely to be the background or interference. Then, the prototype reconstruction is computed with the scores $S_{i,j}^{n}$ as:(12)$$p_{n}^{T2B}=\left[S_{1,1}^{n},\;S_{1,2}^{n},\;\ldots ,\;S_{H,W}^{n}\right]{\left[f_{1,1},\;f_{1,2},\;\ldots ,f_{H,W}\right]}^{T}$$

To reduce the interference and background and refine the prototype and candidate features, the target-to-background enhancement is implemented as:(13)$$f_{i,j}^{T2B}=\left[C_{i,j}^{1},\;C_{i,j}^{2},\;\ldots ,\;C_{i,j}^{n}\right]{\left[p_{1}^{T2B},\;p_{2}^{T2B},\;\ldots ,\;p_{n}^{T2B}\right]}^{T}$$

During the end-to-end training, the enhance feature is refined to be more attentional to reduce interference for better performance.

To make full use of context information and intra-class characteristics, the class-to-class attention and the target-to-background attention are combined for training:(14)$$f_{i,j}^{AFE}=f_{i,j}+\alpha f_{i,j}^{C2C}+\beta f_{i,j}^{T2B}$$
where α and β represent the hyperparameter and are set according to the experiment. The enhanced features $f_{i,j}^{AFE}$ are fed into the predictor for semantic segmentation.

### 3.3. Training and Testing of APMNet

The training phase of APMNet includes two stages: class-prototype regression and Attentional Feature-Enhancement. First, pretrain the feature-extraction module to ensure that the candidate features have high-level semantics for prototype-regression. The labeled data are used to classify the candidates and compute accurate prototypes. Then, regress the prototypes according to the steps included in Section 3.1. The Attentional Feature-Enhancement depends on the regressed prototypes, which are fixed after the regression. During this process, the regressed prototypes are frozen and used as class templates for model updating. during the test phase, the optimized prototypes and the attentional enhancements are used to deal with inference. The detailed descriptions of the training and testing steps are shown in Algorithm 1, and the visualization of the main steps in training and testing APMNet is shown in Figure 2.
**Algorithm 1:** The training and testing steps of the APMNet**Input**: train set $Data_{train}$ and test set $Data_{test}$
**Output**: The segmentation results preds of test set $Data_{test}$
**Training:***Class Prototype Regression***For**$(I_{in},\;gt)\in Data_{train}$**do:** 1. Froze the feature extractor (pre-trained on ImageNet). 2. Feed the $I_{in}$ into the extractor, for the features $f_{i,j}$. 3. Randomly initialize the class prototype proto, and compute the matrix $D_{i,j}$ according to Equation (3). 4. Compute the loss of the Class Prototype Regression $loss_{p}$ according to Equation (7), and start the iterative regression of the proto.**End For***Attentional model update***For**$(I_{in},\;gt)\in Data_{train}$**do:** 1. Unfroze the feature extractor and froze the proto. 2. Feed the $I_{in}$ into the pre-trained feature extractor for the $f_{i,j}$. 3. Compute the attentional enhanced feature $f_{i,j}^{AFE}$ according to Equations (8)–(14). 4. Feed the $f_{i,j}^{AFE}$ into the Predictor and compute the Cross-Entropy loss between the predictions and the gt. 5. Iteratively optimize the feature extractor with the proto frozen.**End For****Testing:****For**$I_{in}\in Data_{train}$**do:** 1. Froze both of the extractor and the proto.  2. Feed the $I_{in}$ into the extractor for the $f_{i,j}$. 3. Compute the attentional enhanced feature $f_{i,j}^{AFE}$ according to Equations (8)–(14). 4. Feed the $f_{i,j}^{AFE}$ into the Predictor for the preds.**End For**

## 4. Experiments

### 4.1. Experimental Setup

In this section, we discuss a set of experiments that we designed to validate the effectiveness and performance of the proposed APMNet in road-scene-parsing tasks. The main datasets involved were CamVid and Cityscapes.

The CamVid dataset is a road-scene dataset consisting of 701 labeled images with a size of 720 × 960, of which 367 images are to be used for training, and 233 images are to be used for testing. The dataset includes 11 labels for common objects.

The Cityscapes dataset consists of 5000 fine-annotated road-scene images, with 2975 images for use in training and 1525 for use in testing. All of the images have a size of 1024 × 2048, and they are labeled with 19 common objects.

In the experiments, the PASCAL VOC intersection-over-union metric (IoU) was adopted for evaluation in CamVid and Cityscapes. Additionally, we used an instance-level intersection-over-union metric (iIoU) in the Cityscapes dataset.

**Training Details****:** In the model training process, commonly used optimization settings referred to RPNet [22]. The SGD optimizer was used with a decay of 0.0005, a batch size of 16, and the learning-rate schedule was set as poly policy, where the learning rate is multiplied by: ${\left(1-iter/maxiter\right)}^{power}$, where iter indicates the current iterate number, the maximum iterate number maxiter is set to 40,000, and the power is a hyperparameter and commonly set to 0.9. The learning rate was initialized as 0.0005.

The hyperparameters of the α, β, γ in the loss-function of the Class-Prototype Regression and the Class-to-class Attention were set by fine-tuning the parameters with numerous tests, and they were finally set to 0.03, 0.02, and 0.1.

**Data augmentation:** The random crop and random horizontal-flip strategy was adopted in the data pre-processing. The input images were cropped to the size of 640 × 640 in CamVid, and they were cropped to the size of 1024 × 1024 in Cityscapes.

### 4.2. Ablation Study

The ablation study was implemented to validate the effectiveness of the modules and algorithms in the Class-Prototype Regression and Attentional Feature-Enhancement. The experiments in this section were based on the Cityscapes dataset. In Section 4.2.1, we adopted ResNet-18 as the feature-extractor to show the effectiveness of the Class-Prototype Regression, while the Section 4.2.2, ResNet-50 was used to demonstrate the performance of the Attentional Feature-Enhancement.

#### 4.2.1. Class-Prototype Regression

The comparison of different strategies in Class-Prototype Regression was based on the methods described in the Section 3.1, as well as the prototype update, the Sample-Selection Algorithm (SSA), and the Class-Repellence Algorithm (CRA). The evaluation of the proposed methods was performed according to the segmentation performance with a setting equal to that of the other part of the model.

The quantitative results are shown in Table 1, and they indicate that, with the gradual optimization of the strategies of Class-Prototype Regression, the performance of prototype-matching became more accurate, and the segmentation result of mIoU improved.

The prototype update helped to update the more-accurate class prototypes, and the features of the class were distributed according to the prototype, which extracts the stable and robust characteristics of a target. However, the imbalanced sample number of classes caused changes in the accuracy of the prototype update, so the Sample-Selection Algorithm reduced the imbalance in the training phase by focusing on the classes with fewer samples or difficult samples. The Class-Repellence Algorithm enhances the target features by amplifying the differences in intra-classes and by reducing the interference. A visual demonstration of the proposed methods is shown in Figure 3.

The colored points represent the prototypes visualized with UMAP [27] tools in 2D space in Figure 3, and Figure 3a–d indicates the results obtained with settings corresponding to those in Table 1. The distribution of the visualized points shows the relations among the regressed prototypes. Updating the prototypes from the dataset simply regresses the feature that represents the classes, where the classes with similar appearances have closer prototypes in Figure 3a. Adding the Sample-Selection Algorithm (SSA) causes the colored points to disperse, achieving a higher level of segmentation accuracy according to Table 1, which means that the SSA helps improve the accuracy of the prototype regression. In Figure 3c, the greater number of dispersed colored points indicates that the Class-Repellence Algorithm (CRA) amplified the difference among classes greatly. Figure 3d shows that the combination of the SSA and CRA enhances both the accuracy and distinguishability of the prototypes, achieving an improvement of 5.6% as compared with the baseline.

Table 2 shows the effects of setting the threshold ε in the Sample-Selection Algorithm to various values to optimize the Class-Prototype Regression. According to Section 3.1.1, ε is used to sample the difficult examples in the regression. If ε is set lower, the sample range becomes smaller, while setting ε higher will limit the prototype’s accuracy. The experiments showed that, when the ε was set to 0.010, the performance was better than it was with the other settings.

#### 4.2.2. Attentional Feature-Enhancement

The comparison in this section is based on the methods described in Section 3.2. The results are shown in the Table 3.

As shown in Table 3, both C2C and the T2B enhance the performance of the model, and their combination achieves a 1.98% improvement as compared with the baseline. C2C focuses on the relationships among the classes and refines the features for attentional enhancement, while T2B makes connections between prototype features and background features and enhances the candidate features based on the target inclination.

APMNet adds the Attentional Feature-Enhancement modules (C2C and T2B) during the testing time. Although the processing time increases, the overall FPS values are acceptable in road-scene applications. The C2C module implements the normalization according to the prototype dimension, while the T2B module implements the normalization according to the spatial dimension. So, the processing time of the T2B module is easily affected by the resolution of the input images and has more computational complexity than the C2C module.

Figure 4 visualizes the enhancement of the Attentional Feature-Enhancement based on Class-to-Class Attention (C2C) and Target-to-Background Attention (T2B). The highlighted areas represent the candidate features to be enhanced. Pixels of all classes are to be enhanced, and for simplicity, the classes of roads, buildings, poles, signs, pedestrians, and cars are shown for demonstration. In Figure 4a, the attention response of the specific class is mainly concentrated on the corresponding area, and the attention responses of the targets are also unevenly distributed according to the strength of the ground visual features. For indistinguishable areas, such as target edges and small targets (poles and signs), there are responses in each class of attention results. The C2C enhancement combines various attention results to enhance strong matching areas and suppress weak matching areas, which are more conducive to feature discrimination. The poles have small, weak visual features with a lot of interference. After the C2C enhancement, the pole features are corrected, and the segmentation results are improved. Road signs are also small targets which are easily obfuscated in complex backgrounds and are thus difficult to detect. Benefits of the C2C enhancement include the amplification of the differences in the characteristics of each class of targets, the enhancement of the separability of similar objects, and heightening of the response of the signs in the attention results.

Cars and pedestrians are moving targets, and near and far cars and pedestrians show different target characteristics: far cars and pedestrians are weak targets with weak visual features, making them difficult to detect; near cars and pedestrians are larger in size in the images, but details and difficult-to-detect areas are also magnified, such as car windows, which often reflect complex textures, and pedestrians’ clothing and accessories, which often interfere in pedestrian-target segmentation. This situation is also reflected in the attention results of relevant regions. The attention drop of the complex areas in the target part will affect the detection effect. With the C2C enhancement, the detection of vehicles and pedestrians can be well adapted to various changes.

Different from C2C enhancement, T2B enhancement pays more attention to the distribution of classes in space, and areas with weak spatial attention-response are suppressed. The C2C enhancement considers the relationship between each target and the class prototype, which can be regarded as an extension of a classifier, while the T2B enhancement considers the spatial-target relationship, and a target with strong confidence will suppress targets with low confidence.

As shown in Figure 4b, the response of spatial attention to non-targets of this type is much weaker than that to targets of this type, which reduces the interference, e.g., the responses of the roads and buildings are significantly reduced on attention maps of other classes. The weak and small targets, such as poles and signs, are also effectively attenuated on other attention maps, but at the same time, the responses of some difficult areas are also attenuated. Therefore, combining these two attention-enhancement methods can expand well and compensate for the insufficiency of the classifier. From the perspective of the final segmentation effect, the segmentation method based on attention-enhancement further improves the feature-separability based on the baseline algorithm. A better road to semantic-segmentation effects has been achieved.

### 4.3. Results with the CamVid Dataset

Due to the small scale of the CamVid dataset, the image size and field of view are small, and a network with too much capacity will lead to overfitting. Referring to the baseline algorithm BiSeNet, this section discusses the use of the ResNet-18 backbone network, which is the same backbone network used by BiSeNet in this experiment, as the feature-extractor in the CamVid dataset experiment.

Table 4 shows the comparison of the accuracy of each algorithm with the CamVid dataset. APMNet had obvious advantages regarding various types of accuracy and total average accuracy. Compared with the baseline algorithm BiSeNet, the improvement of APMNet was almost comprehensive. The class-prototyping-regression strategy adopted by APMNet enabled the detector to greatly improve with the class of difficult samples, such as signs and bicycles. The Attentional Feature-Enhancement also helped APMNet achieve satisfactory results on moving/changing targets such as cars, pedestrians, etc.

Figure 5 shows the visualization results of APMNet and the baseline algorithm on the CamVid dataset. As shown in the figure, APMNet made a significant improvement over the baseline algorithm, both in terms of overall structure and in the details of the target. On small targets, such as poles and traffic lights, the performance of BiSeNet became poorer. In addition, BiSeNet did not offer a good level of robustness for background interference around a target, and false detections often occurred on similar classes of targets, such as the false detection of antennas on buildings as poles. Although sidewalks and lanes are different in spatial position, BiSeNet only discriminated using visual features, which also led to false detections.

The detection effect of APMNet with the addition of the Class-Prototype Regression and Attentional Feature-Enhancement was significantly improved. In APMNet, the classification method of prototype-feature-matching replaced the method of the convolutional classifier. Combined with the Class-Prototype Regression, the difference between the prototypes of each class was enlarged, and the separability became stronger. As shown in the figure, in the APMNet results, the misdetection of complex textures on buildings as poles was minimized, and the accuracy with small objects also increased. Furthermore, with the enhanced feature, APMNet also achieved better performance on the structural completeness. For example, the detected pedestrians were more detailed than with BiSeNet, and there were fewer false detections.

### 4.4. Results with the Cityscapes Dataset

Compared with the CamVid dataset, the Cityscapes dataset is much larger in terms of data scale, image size, and field of view, and the stronger extractor was capable of adapting to the variance in the dataset. In this experiment, ResNet-101 was adopted as the feature-extractor, with the baseline being BiSeNet.

Table 5 shows the accuracy comparison of each algorithm on the Cityscapes dataset. CRPNet had advantages over comparison algorithms. The baseline algorithm BiSeNet only provided the results of the average intersection ratio. On this indicator, CRPNet improved the accuracy by 2% by using enhanced feature-representation. Most of the comparison algorithms are optimized for the feature-extractor of the deep network. Among them, AAF proposes an adaptive affinity field to optimize the segmentation task, and it uses the local constraints of the target pixels to improve the original pixel-level constraints so that the network achieves high-quality, local-structure information. APMNet further uses the C2C and T2B attention mechanism to improve the structure of the prediction results, which improves the performance of CRPNet. Compared to other networks optimized for feature extractors, CRPNet also has great advantages. In addition, due to the prediction method of class-prototype comparison, CRPNet also has a great advantage in the index of the average instance-intersection ratio.

Figure 6 presents the visualization of the results of APMNet and the baseline algorithm with the Cityscapes dataset. As shown in the figure, the baseline algorithm BiSeNet demonstrated poorer performance on small targets, especially with complex backgrounds. In contrast, in the case of APMNet, the detection results were significantly improved through class-prototype enhancement and attention enhancement. Also, the completeness of detected objects was improved. In the detection of poles, due to the fineness of the target itself, the results of BiSeNet often appeared discontinuous, and false detections often occurred in areas where visual features similar to poles appeared at the edge of buildings. In contrast, in the detection results of APMNet, the detection results of poles were closer to the manual-labeling results, and there were fewer false detections in complex interference areas.

## 5. Conclusions

This paper proposes a novel scene-parsing network, Attentional Prototype-Matching Network (APMNet), to enhance the performance of the semantic segmentation in complex and changeable, real-world road scenes. APMNet develops new algorithms for prototype regression, and segments targets by attentional-matching candidate features. The experiments show the great advantages of APMNet, especially for small and weak targets, e.g., poles and signs. Benefits of the prototype-matching methods include the fact that targets with strong interferences still have discernable features. Attention Feature-Enhancement provides more-comprehensive scene information and a greater level of integrity for the segmentations.

Although APMNet boosts performance, the training process is relatively complex, and the computational complexity involved in the Attention Feature-Enhancement is great. Future work will aim to simplify the training steps and improve the efficiency of the training, specifically by integrating the processes of prototype-regression and feature-enhancement, and by developing highly efficient operators of the feature relation.

## Figures and Tables

**Figure 1 sensors-22-06159-f001:**
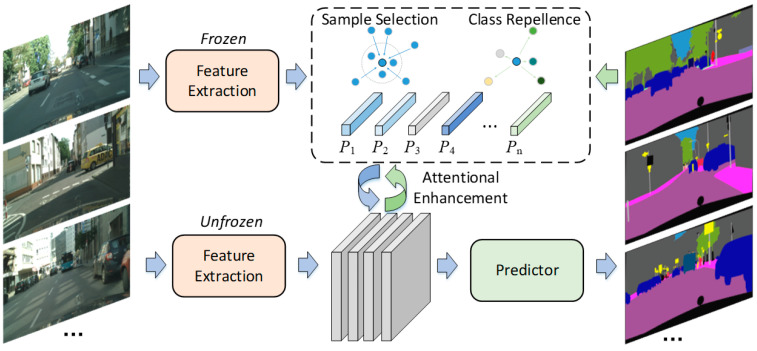
The flowchart of APMNet.

**Figure 2 sensors-22-06159-f002:**
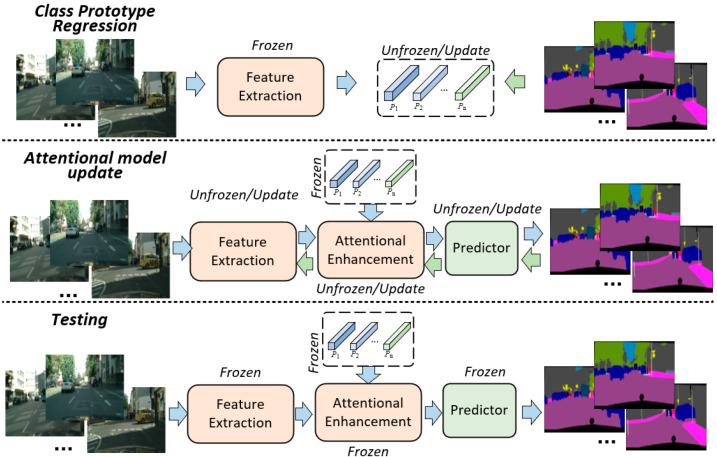
The visualization of the main steps in training and testing APMNet.

**Figure 3 sensors-22-06159-f003:**
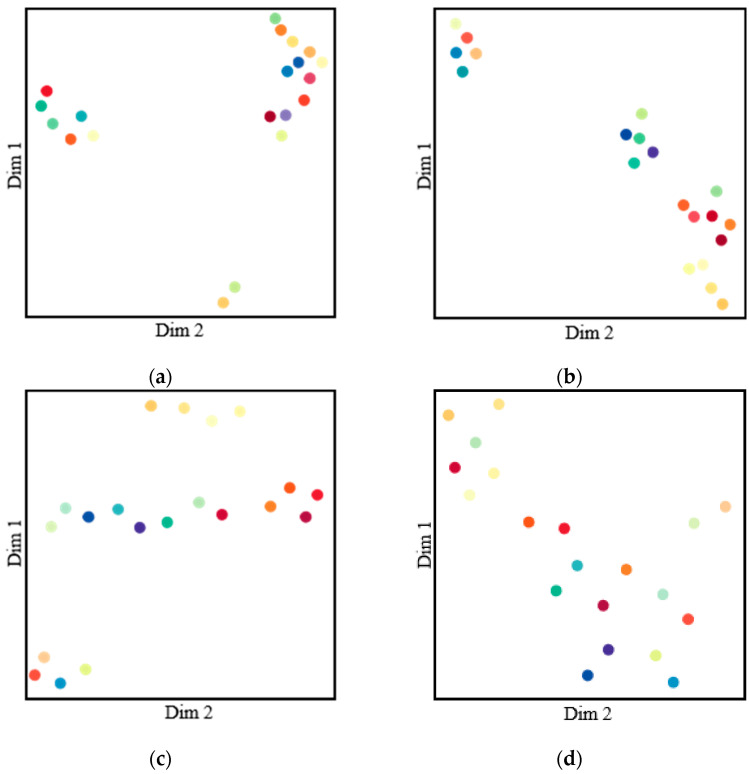
The visualization of class prototypes. (**a**–**d**) in the figure indicate the corresponding settings according to Table 1.

**Figure 4 sensors-22-06159-f004:**
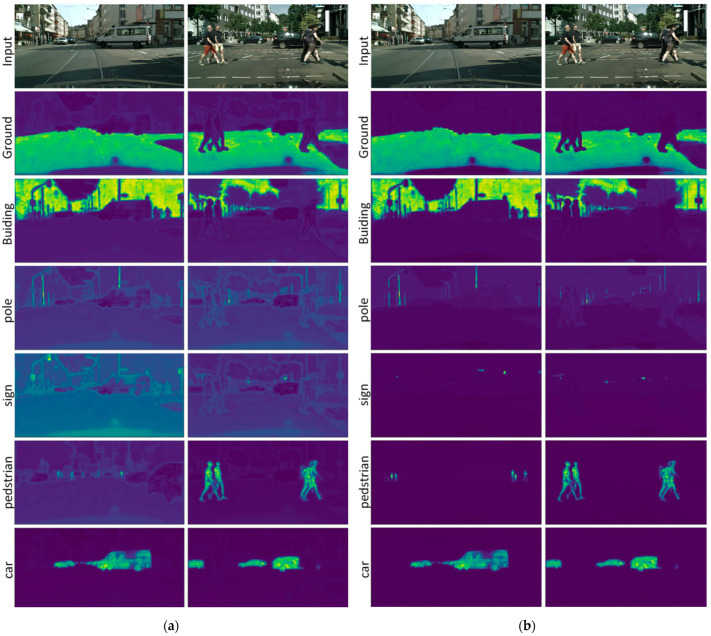
The visualization of the Class-to-Class Attention and the Target-to-Background Attention. (**a**) C2C Attention, (**b**) T2B Attention.

**Figure 5 sensors-22-06159-f005:**
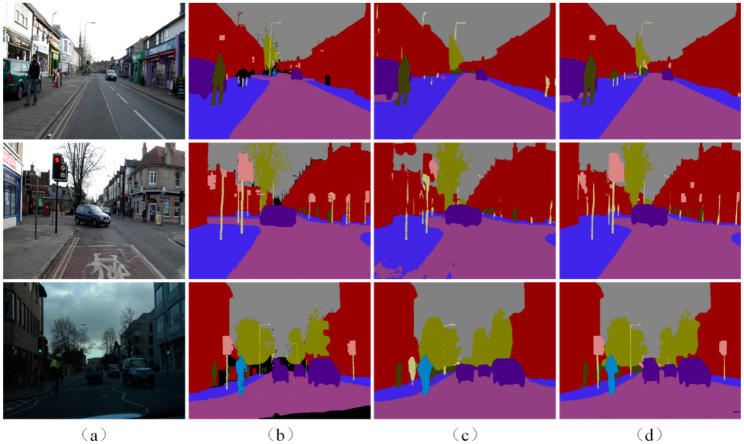
The visualization of the results of APMNet and the baseline with the CamVid dataset: (**a**) Inputs; (**b**) Ground Truth; (**c**) BiSeNet results; (**d**) APMNet results.

**Figure 6 sensors-22-06159-f006:**
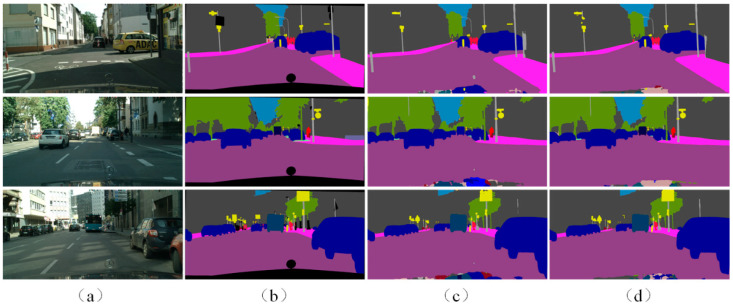
The visualization of the results of APMNet and the baseline on the Cityscapes dataset: (**a**) Inputs; (**b**) Ground Truth; (**a**) BiSeNet results; (**d**) APMNet results.

**Table 1 sensors-22-06159-t001:** Comparison of different settings in Class-Prototype Regression.

	PU	SSA	CRA	mIoU
-				61.4
(a)	✓			65.3
(b)	✓	✓		65.8
(c)	✓		✓	65.7
(d)	✓	✓	✓	67.0

**Table 2 sensors-22-06159-t002:** Comparison of different settings in the Sample-Selection Algorithm.

Ε	mIoU
0.001	65.1
0.005	65.8
0.010	67.0
0.050	66.3
0.100	65.9

**Table 3 sensors-22-06159-t003:** Comparison of the Attentional Features.

C2C	T2B	mIoU	FPS
		74.71	65.5
✓		76.23	41.9
	✓	75.17	37.8
✓	✓	78.88	28.5

**Table 4 sensors-22-06159-t004:** Results with CamVid; “-” indicates that the results were not given in the original paper. The bold results indicate the best for the classes.

Algorithms	Building	Tree	Sky	Car	Sign/Symbol	Road	Pedestrian	Fence	Pole	Sidewalk	Bicycle	mIoU
SegNet-A [7]	75.0	84.6	91.2	82.7	36.9	93.3	55.0	47.5	**44.8**	74.1	16.0	-
SegNet-B	**88.8**	**87.3**	92.4	82.1	20.5	**97.2**	57.1	49.3	27.5	84.4	30.7	55.6
ENet [28]	74.7	77.8	**95.1**	82.4	51.0	95.1	67.2	51.7	35.4	**86.7**	34.1	51.3
BiSeNet-A [17]	82.2	74.4	91.9	80.8	42.8	93.3	53.8	49.7	25.4	77.3	50.0	65.6
BiSeNet-B	83.0	75.8	92.0	83.7	46.5	94.6	58.8	**53.6**	31.9	81.4	54.0	68.7
DeepLab [8]	81.5	74.6	89.0	82.2	42.3	92.2	48.4	27.2	14.3	75.4	50.0	61.6
Dilation8 [29]	82.6	76.2	89.9	84.0	46.9	92.2	56.3	35.8	23.4	75.3	55.5	65.3
APMNet	86.5	78.8	92.5	**89.9**	**59.5**	96.3	**70.6**	44.4	39.5	86.5	**67.6**	**73.8**

**Table 5 sensors-22-06159-t005:** Results with Cityscapes; “-” indicates that the results were not given in the original paper. The bold results indicate the best for the classes.

Algorithms	IoU Class	iIoU Class	IoU Category	iIoU Category
DPN [30]	66.8	39.1	86.0	69.1
DeepLab [8]	70.4	42.6	86.4	67.7
LDFNet [31]	71.3	46.3	88.5	74.2
GLR [32]	77.3	53.4	90.0	76.8
PSPNet [9]	78.4	56.7	90.6	78.6
BiSeNet [17]	78.9	-	-	-
AAF [13]	79.1	56.1	90.8	78.5
LDN [33]	79.3	54.7	90.7	78.4
CFNet [19]	79.6	-	-	-
DFN [34]	80.3	58.3	90.8	79.6
HANet [35]	80.9	58.6	91.2	79.5
APMNet	**80.9**	**60.2**	**91.6**	**80.5**

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
