# Peer review of "Road-Scene Parsing Based on Attentional Prototype-Matching"

_sensors, 2022, doi:10.3390/s22166159_

Round 1
Reviewer 1 Report
The authors of this paper claim that the road scene is complex and b background interferences will ruin the visual structure of the image to detect the target object. Therefore, the authors of this paper propose a novel scene parsing network known as the Attentional Prototype Matching Network (APMNet), which can segment targets by matching candidate features with target prototypes that have been regressed from labelled road scene data. The authors design the Sample Selection and the Class Repellence Algorithm and build the Class-to-Class and Target-to-Background Attention to increase the feature recognizability based on the target visual characteristics and spatial target distribution. This paper is easy to follow and well organized. It would be better if the authors revised the paper based on the following comments.
Comments
1) In the section on performance evaluation, the authors should consider including a discussion on hyperparameter optimization for optimized performance.
2) The authors should consider including a comparison of the proposed model's processing time to that of other existing models in the section on performance evaluation.
Author Response
1) In the section on performance evaluation, the authors should consider including a discussion on hyperparameter optimization for optimized performance.
Response: Thanks for your comments, and we have added Table 2 to discuss different settings of the hyperparameter ε in the Sample Selection Algorithm. According to section 3.1.1, the ε is used to sample the hard examples in the regression. If the ε is set lower, the sample range gets small, while the ε is set higher, and the prototype accuracy will be limited. The experiments show that when the ε is set to 0.010, the performance is higher than the other settings.
The hyperparameters of the α, β, and γ in the loss function of the Class Prototype Regression and Class-to-class Attention are set by fine-tuning the parameters with numerous tests, and finally set to 0.03, 0.02, and 0.1.
2) The authors should consider including a comparison of the proposed model's processing time to that of other existing models in the section on performance evaluation.
Response: Thanks for your comments, and we have added FPS data in the Table 3, which shows processing time comparison of different settings of the APMNet and the baseline BiSeNet. Although the inference time increases, the overall FPS values of the APMNet are affordable in road scene applications.
In the benchmarks of CamVid and Cityscapes, the reports of the candidate methods commonly contain multiple tricks, such as multi-scale and flip inference, which significantly decrease the efficiency in the road scene applications. So, the processing time is not discussed in the experiments on the CamVid and Cityscapes datasets.

Reviewer 2 Report
***************** Summary: ******************
The authors motivate that road scene parsing is subject to difficulties as road scenes are complex, constantly changing and background interference is likely. They propose the "Attentional Prototype Matching Network (APMNet) to partition road scenes by matching candidate features to known and labelled target prototypes. They compare an implementation of their approach with other state of the art approaches.
***************** Decision: *****************
To this reviewers best knowledge, the topic of the paper addresses an important issue in the field of image parsing. The results from the experiments in Sect. 4 also indicate a visually comprehensible improvement, compared with some other approaches. However, from the introduction and the related work section, it is not completely clear, what the main highlights of this approach are. Also, the formalisation in Sect. 3 lacks motivation and formal details. Without knowing the mathematics of symbols, it is difficult to understand the formalisations that include them. Some inconsistencies and difficult to parse sentences also indicate that this paper is not mature enough for publication and needs more reworking than I see it being possible within a major revision. However, with some distinct improvements according to reviewer comments, this paper might very well be acceptable for publication in another issue of this (or another) journal.
******** Language and presentation: *********
* The language needs improvement. I attached a marked pdf document that should be carefully taken into account for the paper improvement. The markings are indicating minor language issues, while I discuss more detailed, major, issues in the remainder of this review.
* Some strange spacing errors occured in the paper title, as well as in most section headers. Some typos and inconsistencies were also found, also in some formulae (see comments below).
*************** Major issues: ***************
* Related Work/ Embeddedness into the field: From the introduction and the related work section, it does not get completely clear, what the core contribution is. It is somewhat understood from the third paragraph on p.2, that the two step procedure of APMNet, as well as the special feature enhancement methods, are the contribution. But it is difficult to relate the three items that are marked as "highlights" on p.2 below this to that contribution.
* p.4, l.147: Figure 1 needs to be described. What does the reader see in that Figure?
* Not many preliminaries are provided, but section 3.1 starts with using several, formal concepts. Although I understand that they may be well-known within the community, they should still be motivated each with 1-3 additional sentences. E.g., what is a "prototype feature" p_n, "features map" f_{i,j}, "feature dimension", "predictor",... To be able to understand all the calculations and formulae that are contained within Sect. 3, the reader needs to be able to understand the formats/domains of these concepts.
* p.4. l.160: Something seems to be wrong in formula (2). Both the expressions before "* f_{i,j}" and before "* 1" are exactly the same. With this, The formula can be reduced to "f_{i,j} / 1", which basically leads to "P_n = f_{i,j}", which does not make sense.
* p.6. l.229, Sect. 3.3: The training and testing phase of APMNet needs more description than this one paragraph. For the here presented Algorithm 1, it would be nice to see an exemplary execution of the algorithm here. Is it possible to describe the steps using an example image, e.g. like it is done later with Figure 3? If not, this algorithm at least needs some more description.
* Conclusion: This needs to be extended. It is not enough to say that "The experiments show the great advantages of the APMNet.". What exactly are these advantages? Similarly, "The future work is aimed to simplify the training steps and improve the efficiency of the training." needs to be expanded. How exactly could such a simplification work?
*************** Detailed issues: ***************
* General: Mostly the authors name their approach by "Attentional Prototype Matching (APMNet)" (e.g. p.1, l.12, p.2, l.87, p.3, l.139) and only in the beginning by "Attentional Target Prototype Matching Network (ATPM-Net)" (e.g. p.2, l. 71, 72, 79). As far as I understood, with both names, the same approach is meant.
* p.1, l.37: "design encoder-decoder network structure fuse different levels of features" --> I cannot parse this. Possibly it is "for" instead of "fuse"? With that it makes sense.
* p.1, l.39: "These practices enrich receptive information for the features" --> for which features?
* p.2, l.52: "However, the receptive field in the adaptive affinity field is pixel level, and the improvement is limited." --> Why exactly is the improvement limited, because of the pixel level? Please give one explanatory sentence.
* p.2. l. 77: "this chapter uses spatial and class attention feature enhancement methods to enhance the segmentation performance of the network." --> what chapter? I think that another word is meant. Maybe "approach"? (same holds for the occurence of "chapter" in line 83)
* p.2, l.89 ff.: "The build the Class-to-Class and Target-to-Background Attention to increase the feature recognizability based on the target visual characteristics and spatial target distribution." --> first word quite probably should be a "we". Major problem with that sentence: It has not been motivated before what "Class-to-Class and Target-to-Background Attention" are. So that this bullet point of the contribution is understandable, you need to motivate these words before this. The same holds for the "Samples Selection and the Class Repellence Algorithm" from the previous sentence. For the reader to be able to understand your highlights, you need to motivate them.
* p.3, l.116: "The prototype-based introduce the metric leaning to the recognition tasks. Deep-NCM cumulate mean of the embeddings of the classes’ samples to transform classification to a target matching task[15]." --> The first sentence seems incomplete and does not make sense.
* p.3, l.125: "The prototype-based models are promising but existing prototypes are obtained are trained or defined prior to training, and the matching process are interfered strongly in complex environment." --> language: possibly "that are obtained"? Understanding: I do not understand what it means that prototypes "are trained or defined prior to training". So the prototypes are trained before training? What does that mean?
* p.3, l.129: "The recent attention mechanisms in segmentation networks improves their representation." --> Representation of what? I cannot understand this sentence. (Also it is "improve" due to the plural)
* p.3., l.134: "In this paper, we reference the attention and [...]" --> What does this mean? Do you mean that you refer to related work on "attention networks"? If yes, please rewrite the sentence.
* p.4, l.150: What is "R^{H x W}"? This has not been introduced.
* p.4, l.158: "where the indicates the class number and the is the prototype feature " --> two variables or formal expressions are missing.
* p.4, l.159: "The feature dimension of pn and fi,j is C. " --> What does "feature dimension" mean? How does C look like? What is contained in it?
* p.4., l.162: "Compute the feature distances [...], between [...] and [...] and upgrade the [...] through the minimization of [...], for accurate the class prototype feature" --> maybe "to make the class prototype feature more accurate" ?
* p.5, l.171: "[...], and the sReLU indicate the selection activation." --> I do not understand what "sReLU (D_{i,j})" might do in formula (4). It seems to be a function. What is the rannge of this function?
* p.5, l.182: "[...] where a indicate the hyperparameters to improve the training, which is set according to the experiments." --> how are these hyperparameters set according to experiments? They are mentioned once in the experiment later on p.7, l.254, but it is not motivated why it is set to 0.9 there.
* p.5, l.191: "The pixel level D^n_{i,j}, actually reflect spatially class distributed possibility in the input images." --> Possibly it is : "The pixel level D^n_{i,j}, actually reflects the spatial class distribution possibility in the input images." (but I am not sure about that) --> still, this sentence is hard to parse. What does it mean? Possibly elaborate more on this.
* p.5, l.192: "chrematistics" --> "characteristics"?
* p.6, l.217: "where [...], indicate the normalized scores of the prototype to the spatial features." --> I cannot parse this. what does "scores to the spatial feature" mean?
* p.7, l.251: some parameters or variable names are missing in this sentence.
* p.9., l.298: "The comparison in Attentional Feature Enhancement is based [...]" --> "in this section" instead of "in Attentional Feature Enhancement"?
* p.10, l.320: "The pole is a small with weak visual features with much interferences." --> "a small" what? a word is missing
* p.11, l.344: "the responses of road and building on attention other maps are significantly reduced." --> difficult to parse
* p.13, l.394: "[...] and the larger-capacity model is capable to the variance in the dataset." --> difficult to parse. Maybe "to show some variance in the dataset" or "to reflect some variance in the dataset"?
* p.13, l.403: "[...], so that the network obtain local structure. " --> Maybe "[...], so that the network gets locally structured. " ? Although I do not have any idea what that would mean.
* p.13, l.412: "In contrast, APMNet through class prototype enhancement and attention enhancement, in these cases, the detection results are significantly improved, and the completeness of detected objects is improved. " --> Maybe something like this is meant: "In contrast, in the case of APMNet, the detection results are significantly improved through class prototype enhancement and attention enhancement. Also, the completeness of detected objects is improved."
*********** List of References **************
* Several issues exist, some of them major:
* Major: Publication in conferences have been assigned wrongly to papers:
* Reference 6.: "Florian, L. and S.H. Adam. Rethinking atrous convolution for semantic image segmentation. in Conference on Computer Vision and Pattern Recognition (CVPR). IEEE/CVF. 2017." --> After extensive research I have found no indication that this paper was published in CVPR/CVF 2017, nor anywhere actually. Instead it only seems to be available on arxiv. There, the authors gave the text "Computer Science - Computer Vision and Pattern Recognition" only as keywords for their paper, which I find highly problematic, as this indicates an official publication that never existed. This should be carefully checked. Also, the paper originally has four authors, 2 of which have been forgotten in the reference and the two other author names have either wrong sur- or pre-names. Of course, this is the result of my research and I might be wrong. If this was indeed published at CVPR/CVF, I would like to get a link to the publication at IEEE in the author response.
* Reference 15.: "Guerriero, S., B. Caputo and T. Mensink. DeepNCM: Deep Nearest Class Mean Classifiers. in International Conference on Learning Representations 2018." --> This seems to have been published not at the main ICLR conference, but in the workshop track. This needs to be in the reference. Authors also should be consistently "Guerriero, S., Caputo, B. and Mensink, T." (or the other way around for all)
* Pre- and sur-names are abbreviated inconsistently. Sometimes the ordering of pre-names and surnames is also different (e.g. "Florian, L. and S.H. Adam")
* Sometimes there are ",", "." or simply nothing between author names and paper titles.
* Sometimes page numbers are provided, sometimes not, even though the respective entry is published in a book/journal/proceedings.
* Many references seem to have been entered manually. I highly recommend to rework the entire references section and use automatically generated reference entries from trustworthy sources. With this, most of the mentioned problems will not be an issue anymore.

Author Response
Major issues
(1) Related Work/Embeddedness into the field: From the introduction and the related work section, it does not get completely clear, what the core contribution is. It is somewhat understood from the third paragraph on p.2, that the two-step procedure of APMNet, as well as the special feature enhancement methods, are the contribution. But it is difficult to relate the three items that are marked as "highlights" on p.2 below this to that contribution.
Response: Thanks for your comments, we have added more descriptions on the concept of “Prototype”, and the motivations of the APMNet, the Prototype Regression, and the Attentional Enhancement.
(2) p.4, l.147: Figure 1 needs to be described. What does the reader see in that Figure?
Response: We have added the description about the Figure 1
Figure 1 shows the two main procedures of the APMNet, the dotted line indicates the Class Prototype Regression, and the cycle arrow indicates the Attentional Feature Enhancement in the training phase. The lower part shows the testing flow path after training.
(3) Not many preliminaries are provided, but section 3.1 starts with using several, formal concepts. Although I understand that they may be well-known within the community, they should still be motivated each with 1-3 additional sentences. E.g., what is a "prototype feature" p_n, "features map" f_{i,j}, "feature dimension", "predictor",... To be able to understand all the calculations and formula that are contained within Sect. 3, the reader needs to be able to understand the formats/domains of these concepts.
Response: We have added the definition of the "prototype feature", and improved the expressions to make it easy to understand.
The prototypes in the APMNet are defined as the features that have the maximum similarity with samples of a specific class, and the predictor is a sequence of convolution layers to convert features to segmentations
(4) p.4. l.160: Something seems to be wrong in formula (2). Both the expressions before "* f_{i,j}" and before "* 1" are exactly the same. With this, The formula can be reduced to "f_{i,j} / 1", which basically leads to "P_n = f_{i,j}", which does not make sense.
Response: We have corrected the mistake:
(5) p.6. l.229, Sect. 3.3: The training and testing phase of APMNet needs more description than this one paragraph. For the here presented Algorithm 1, it would be nice to see an exemplary execution of the algorithm here. Is it possible to describe the steps using an example image, e.g. like it is done later with Figure 3? If not, this algorithm at least needs some more description.
Response: We have added more descriptions as well as visualization to demonstrate the training and testing process of the proposed APMNet.
(6) Conclusion: This needs to be extended. It is not enough to say that "The experiments show the great advantages of the APMNet.". What exactly are these advantages? Similarly, "The future work is aimed to simplify the training steps and improve the efficiency of the training." needs to be expanded. How exactly could such a simplification work?
Response: We have extended the Conclusion and added more detailed descriptions.
Detailed issues
Response: Thank you for your careful revision, and we have modified the manuscript item by item. All revisions are marked up using the “Track Changes”
List of References
Response: We have updated the References.

Reviewer 3 Report
The paper is well written. It contain a sufficient data for experiments, pictures, actual bibliography. Results and conclusions are understandable and meaningful. But unfortunately the paper contain a lot of small mispints. I strongly recommend the authors to read the text once again and correct all misprints.
These are only some of them:
1. Abstract, 2nd sentence: The Key to... ("Key" with the capital)
2. Subsection 2.2, 2nd sentence: ... of the classes' samples... (please remove the apostrophe)
3. Subsection 4.2.2, 1st sentence: ...Attention(T2B).the results... (please remove dot before ".")
4. The Introduction, page 2, 3rd paragraph: ...Network (ATPM-Net) for. (please complete the sentence correctly).
5. And so on.
Please read once again and correct misprints.
Author Response
Thank you for your careful revision, and we have modified the manuscript item by item. All revisions are marked up using the “Track Changes”
Reviewer 4 Report
The paper is definitely not ready for publication as it is poorly written without taking care of the language rules. Starting from the abstract, it is extremely difficult to understand the sentence as they are written disregarding grammatical rules. Therefore, the paper first must be carefully revised taking into consideration the language requirements.
Author Response
Thank you for your comments, and we have carefully revised and polished the manuscript. All revisions are marked up using the “Track Changes”
Round 2
Reviewer 1 Report
The revised paper has addressed my concerns. The revision is satisfactory, and I am satisfied with the current version.
Reviewer 2 Report
As far as I can see, most of my comments have been addressed by the authors very carefully. The paper has been significantly improved.
The added description of formalities help a lot with understanding the paper. This reader still has some issues with understanding some of the technicality. However, as I am not an expert in this field, I will follow the decision of the more expert reviewers who confirmed the soundness of the approach.
Minor remark:
p.1, abstract: "The road scene parsing a complex and changeable, the interferences in the background destroys [...]" --> "The road scene parsing is complex and changeable, the interferences in the background destroy [...]" (change "a" with "is" and remove the s from "destroys") (perhaps this new error occured due to my badly readable handwriting in the previously attached pdf, apologies for that!)
Reviewer 4 Report
The authors have responded to all my queries, therefore, it may be accepted in its present form.